# Influence of Rigid–Elastic Artery Wall of Carotid and Coronary Stenosis on Hemodynamics

**DOI:** 10.3390/bioengineering9110708

**Published:** 2022-11-18

**Authors:** Muhamed Albadawi, Yasser Abuouf, Samir Elsagheer, Hidetoshi Sekiguchi, Shinichi Ookawara, Mahmoud Ahmed

**Affiliations:** 1Department of Energy Resources Engineering, Egypt-Japan University of Science and Technology (E-JUST), P.O. Box 179, New Borg El-Arab City 5221241, Egypt; 2Biomedical Flow Dynamics Laboratory, Institute of Fluid Science, Tohoku University, Sendai 980-8577, Japan; 3Engineering Mathematics and Physics Department, Faculty of Engineering, Alexandria University, Alexandria 5424041, Egypt; 4Mechanical Engineering Department, Faculty of Engineering, Alexandria University, Alexandria 5424041, Egypt; 5Faculty of Engineering, Aswan University, Aswan 81528, Egypt; 6Department of Chemical Engineering, Graduate School of Science and Engineering, Tokyo Institute of Technology, Ookayama, Meguro-ku, Tokyo 152-8552, Japan; 7Mechanical Engineering Department, Assiut University, Assiut 71516, Egypt

**Keywords:** FSI, CFD, stenosis, coronary artery disease (CAD), cardiovascular diseases

## Abstract

Cardiovascular system abnormalities can result in serious health complications. By using the fluid–structure interaction (FSI) procedure, a comprehensive realistic approach can be employed to accurately investigate blood flow coupled with arterial wall response. The hemodynamics was investigated in both the coronary and carotid arteries based on the arterial wall response. The hemodynamics was estimated based on the numerical simulation of a comprehensive three-dimensional non-Newtonian blood flow model in elastic and rigid arteries. For stenotic right coronary artery (RCA), it was found that the maximum value of wall shear stress (WSS) for the FSI case is higher than the rigid wall. On the other hand, for the stenotic carotid artery (CA), it was found that the maximum value of WSS for the FSI case is lower than the rigid wall. Moreover, at the peak systole of the cardiac cycle (0.38 s), the maximum percentage of arterial wall deformation was found to be 1.9%. On the other hand, for the stenotic carotid artery, the maximum percentage of arterial wall deformation was found to be 0.46%. A comparison between FSI results and those obtained by rigid wall arteries is carried out. Findings indicate slight differences in results for large-diameter arteries such as the carotid artery. Accordingly, the rigid wall assumption is plausible in flow modeling for relatively large diameters such as the carotid artery. Additionally, the FSI approach is essential in flow modeling in small diameters.

## 1. Introduction

The cardiovascular system warrants blood to transport oxygen and nutrients through the body. The heart’s left side receives and pumps the oxygenated blood to the rest of the body through the aorta. The cardiovascular system includes the aorta, coronary, carotid arteries, and others. The coronary arteries emerge from the ascending aorta and supply blood to the heart. Small coronary branches go through the heart muscle to provide it with the necessary oxygenated blood. Additionally, the neck’s main blood vessels, the carotid arteries, carry blood to the brain, neck, and face. Every carotid artery in the neck splits into two branches: the external carotid artery (ECA), which delivers blood to the face and neck, and the internal carotid artery (ICA), which supplies blood to the brain [1].

Cardiovascular disease (CVD) is collectively referred to as a condition affecting the heart and the blood vessels. It’s usually characterized by the thickening and hardening of the vessel walls due to endothelial dysfunction and plaque development, defined as atherosclerosis. Atherosclerosis is a significant arterial disease associated with the accumulation of cholesterol and other lipids beneath the internal layer of the artery and causes stenosis, which is a reduction in the cross-sectional area of the lumen [2].

Endothelial dysfunction is a primary precursor to atherosclerosis leading to a disturbance in hemodynamics [3]. The plaques accumulate over distorted endothelial cell regions due to low wall shear stress and associated descriptors (flow circulation) [4]. Low wall shear stress (WSS) leads to plaque susceptibility and initiation. Once the plaques are formed, an increase in WSS value is observed. Accordingly, high WSS leads to a rupture of the plaques and potentially even thrombosis [5]. Eshtehardi et al. [6] epitomized the clinical data associating high WSS with prospective endothelial cell damage as a putative etiological mechanism underpinning high-risk plaque formation. Endothelial shear stress is associated with plaque development characteristics with low defined as <1 Pa, physiologic (intermediate) of 1–2.5 Pa, and high WSS >2.5 Pa [7,8].

Understanding the mechanisms underlying the initiation and progression of atherosclerosis requires a thorough knowledge of blood flow. Several clinical methods have been used for the in vivo investigation of blood flow-related variables, such as the use of phase contrast magnetic resonance imaging (MRI), Doppler ultrasound, and particle-based methods such as particle image velocimetry (PIV) [9,10]. Doppler ultrasound can be used for early diagnosis of internal carotid artery (ICA) stenosis through extracranial hemodynamics [11]. Thereby, computational simulation emerges as a more efficient alternative to predict blood behavior and hemodynamics [12]. The hemodynamics was numerically investigated in symmetrical and asymmetrical bifurcating of pulsatile flow in simplified geometries [13]. Mekheimer et al. [14] presented numerically that mixing the blood with the synovial fluid can change the rheological properties of the blood and the mechanical characteristics of the formed stenosis. Accordingly, computational fluid dynamics (CFD) has been extensively used in the investigation of hemodynamics [15]. Additionally, CFD can lead to non-invasive procedures for diagnosing different diseases, such as atherosclerosis before they proceed to severe instances [16,17]. Taebi [18] presented recent deep-learning approaches integrated with CFD for computational hemodynamics.

The vascular arterial wall has an extremely intricate structure with various mechanical properties. The derivation of accurate comprehensive models for such a complicated structure is highly difficult and continues to be a challenging point of active research. High-complexity models are essential in capturing detailed features of the material’s mechanical behavior. However, simpler models are less accurate but more practicable from mathematical and computational standpoints. Accordingly, a proper balance between high-complexity and simpler models is necessary to obtain the mechanical features with appropriate computational simulation.

The arterial behavior was approximated to be a rigid body using fluid flow modeling [19]. Further studies considered fluid domain simulations coupled with finite element analysis of the arterial wall response through two-way fluid–structure interaction (FSI). FSI is a multiphysics coupling of fluid dynamics and structural mechanics regulations. This phenomenon, which can be steady or oscillatory, is characterized by interactions between a deformable or moving structure and a surrounding or interior fluid flow. This multiphysics coupling is a more realistic approach for simulating the influence of the blood flow on the artery vessel and vice versa [20]. However, FSI necessitates additional modeling assumptions regarding the vessel’s mechanical properties and significantly more computational effort. Several researchers investigated the hemodynamics of arteriosclerosis diseases using the FSI procedure in stenosed vessels [21,22]. Dong et al. [23] presented a correlation between the angulation of the coronary artery branches and the local mechanical and hemodynamic stresses at the artery bifurcation using FSI analysis. Additionally, Failer et al. [24] investigated the impact of using FSI to simulate blood flow in simple stenotic geometry. Zouggari et al. [25] investigated the influence of plaques on the WSS distribution using FSI analysis and CFD simulations. The obtained results showed attenuation of WSS values only at the plaque region.

The morphologic features of plaques can be classified based on how the artery is narrowed as asymmetric (eccentric stenosis) and axisymmetric (concentric stenosis) [26]. Different stenosis morphologies were investigated, including oval, bean-shaped, and crescent, either with or without eccentricity [27]. Eccentric lesions are frequently common in patients with partially occlusive thrombus or ruptured atherosclerotic plaques [28]. Additionally, according to Poepping et al. [29], flow patterns varied amongst stenoses of various eccentricities, with the salient changes reflected in the circulation regions’ size and position.

Based on a review of the literature, several studies investigated the influence of wall compliance on the hemodynamics of stenotic arteries. However, to the authors’ knowledge, no reported research has assessed the rigid wall assumption of patient-specific arteries based on hemodynamics. Therefore, the originality of the current work is to investigate the hemodynamics in coronary and carotid arteries and compare the obtained results with the corresponding values obtained using rigid wall assumption. The hemodynamics was estimated based on the numerical simulation of a comprehensive three-dimensional non-Newtonian blood flow model in elastic and rigid arteries. Accordingly, the two-way FSI approach is considered when coupling the blood flow model with the finite element analysis of wall elasticity in the case of eccentric stenoses with a moderate degree of severity. Lastly, a comparison between predicted results of coronary and carotid arteries with those obtained under rigid wall assumption is reported and discussed.

## 2. Problem Definition and Modeling

### 2.1. Physical Model

In the present study, three-dimensional patient-specific geometries of the fluid domain of the carotid and right coronary arteries are considered, as shown in Figure 1. The anatomic data of two healthy patient-specific geometries were detected with the carotid artery (CA) and right coronary artery (RCA), which were obtained from a clinically indicated computed tomography (CT) angiogram. The CT technology allows non-invasive visualization inside the human body, giving more capability for a more accurate diagnosis. Moreover, the CT provides clear images of the vessels’ anatomy and the presence or absence of vascular disorders. The CT technique develops digital DICOM (Digital Imaging and Communication in Medicine)-formatted two-dimensional images. Each image represents a layer of the arterial vessel. The 3D geometries of the CA and RCA were imported to a trial version of 3-Matic software to reduce the imperfections and abnormalities that resulted from the segmentation process. Moreover, eccentric stenosis is imported to the RCA at 10 mm before the bifurcation (model A), as shown in Figure 1a. Additionally, eccentric stenosis is imported to the healthy geometry of the CA at the internal carotid artery (ICA) 5 mm downstream of the artery bifurcation (model B), as shown in Figure 1b. Additionally, eccentric stenosis is imported to the CA at the common carotid artery (CCA) 8 mm before the bifurcation (model C), as shown in Figure 1c. The selected location for each artery is at the most common potentially stenotic region in the CA and RCA, as reported [30,31]. The selected locations are mainly due to the fact that the cardiovascular system is exposed to atherosclerosis around the bifurcation. At each location, a reduction percentage in hydraulic diameter $\left(\frac{d_{\mathrm{artery}}-d_{\mathrm{throat}}}{d_{\mathrm{artery}}}\times 100\right)$ is 60%, representing moderate stenosis according to the North American Symptomatic Carotid Endarterectomy Trial (NASCET) method [32] and the Society of Cardiovascular Computed Tomography (SCCT) guidelines [33]. Furthermore, the healthy CA has a geometry of 7.82 mm hydraulic diameters for the CCA and 4.52 mm and 6.41 mm hydraulic diameters for ECA and ICA, respectively. Additionally, the healthy RCA has a geometry of 3.77 mm hydraulic diameter for the main artery, 2.4 mm at the outlet (1), and 1.9 mm at the outlet (2). Moreover, the structure domain for the FSI analysis is investigated with a thickness 0.66 mm in the CA geometry [34] and 0.55 mm in the RCA geometry [35]. The physical models’ dimensions are summarized in Table 1.

### 2.2. Theoretical Analysis

Modeling the blood flow in the arterial geometries is three-dimensional, laminar, unsteady, and fluctuating. To predict the hemodynamics of the investigated geometries of the carotid artery and right coronary artery, 3D Navier–Stokes equations are developed based on a non-Newtonian relation between the shear stress and the rate of shear strain. The cellular blood components include leukocytes (white blood cells), which are a part of the immune system; thrombocytes (platelets), which are crucial for blood clotting; and erythrocytes (red blood cells), which carry oxygen and carbon dioxide to and from the organs. About 55% of blood is made up of plasma, and the remaining 45% is made up of erythrocytes. The blood’s leukocyte and thrombocyte populations are incredibly minimal. When erythrocytes are present, the density of the plasma, which is $\;1025\;\mathrm{kg}/m^{3}$, rises to $1050\;\mathrm{kg}/m^{3}$ [1]. Therefore, the following assumptions are considered:

The flow is incompressible.The value of the Reynolds number does not exceed 2000. Accordingly, it is reasonable to assume the flow is laminar. Such an assumption agrees with the previous study of Peacock et al. [36], which stated that the coronary arteries flow’s waveforms were unlikely to be disturbed, and the blood flow in the carotid arteries is usually laminar.Approximately 55% of blood is made up of plasma. As is well known, blood is a non-Newtonian fluid when red blood cells are present, even though plasma is a Newtonian fluid [37].

The vascular walls consist of three layers: adventitia, media, and intima, with various thicknesses and mechanical properties. However, in this study, a linear elastic model has been used with Young’s modulus averaged value equal to 1.08 MPa, Poisson ratio 0.49, and artery wall density $1120\;\mathrm{kg}/m^{3}$ [38].

#### 2.2.1. Governing Equations

The fluid–structure interaction procedure concerns fluid and arterial wall domains, interface, and the relevant boundary states. Modeling the blood flow inside the arterial geometries is three-dimensional, non-Newtonian, laminar, unsteady, and fluctuating.

For a non-Newtonian incompressible fluid, the mass and momentum conservation equations can be expressed as follows:(1)$$\nabla .\vec{V_{f}}=0,\;\;$$
(2)$$\rho_{f}\frac{D\vec{V_{f}}}{Dt}=\rho \vec{g}+\nabla .\tau_{ij},$$
where $V_{f}$ is the fluid velocity vector, $\rho_{f}$ is the fluid density, *g* is the vector acceleration of gravity, and $\tau_{ij}$ is the stress tensor. The body force can be neglected [39]. The stress tensor can be represented as follows:(3)$$\tau_{ij}=\left[\begin{array}{ccc} \tau_{xx} & \tau_{xy} & \tau_{xz}\\ \tau_{yx} & \tau_{yy} & \tau_{yz}\\ \tau_{zx} & \tau_{zy} & \tau_{zz} \end{array}\right],$$

The volume fraction of erythrocytes in plasma is the primary determinant of blood characteristics. The precise composition of the blood is the critical factor to consider when selecting an appropriate non-Newtonian model [40,41]. In the present study, the Carreau model is considered, representing the relation between the dynamic viscosity (μ) and strain rate ($\dot{\gamma }$) as [42]
(4)$$\mu =\mu_{\infty }+\left(\mu_{o}-\mu_{\infty }\right)\times {\left[1+{\left(\lambda \dot{\gamma }\right)}^{2}\right]}^{\frac{n-1}{2}},$$

The coefficients for the Carreau model could be written as follows: the value of zero shear rate viscosity, $\mu_{0}=0.056\;\mathrm{Pa}\cdot s$, the infinite shear rate viscosity, $\mu_{\infty }=0.00345\;\mathrm{Pa}\cdot s$, time constant, λ=3.131 s, and the power index, n=0.3568 [43]. Consequently, the shear stress is expressed as follows:(5)$$\tau_{ij}=-p\delta_{ij}+\left(\mu_{\infty }+\left(\mu_{o}-\mu_{\infty }\right)\times {\left[1+{\left(\lambda \dot{\gamma }\right)}^{2}\right]}^{\frac{n-1}{2}}\right)\left(\frac{\partial u_{i}}{\partial x_{j}}+\frac{\partial u_{j}}{\partial x_{i}}\right),$$

Hemodynamics such as wall shear stress (WSS), time-averaged wall shear stress (TAWSS), and the velocity field are described as the following [44,45].

The WSS can be written as follows:(6)$$\;\tau_{w}{}_{ij}=\left(\mu_{\infty }+\left(\mu_{o}-\mu_{\infty }\right)\times {\left[1+{\left(\lambda \left(\frac{\partial u_{i}}{\partial x_{j}}+\frac{\partial u_{j}}{\partial x_{i}}\right)\right)}^{2}\right]}^{\frac{n-1}{2}}\right)\times \left(\frac{\partial u_{i}}{\partial x_{j}}+\frac{\partial u_{j}}{\partial x_{i}}\right),\;\;\left(i\neq j\right),$$
where $\;\tau_{w}$ is the wall shear stress.

The TAWSS is calculated by integrating WSS magnitude over the cardiac cycle as shown in Equation (7).
(7)$$TAWSS=\frac{1}{T}\int_{0}^{T}\left|\vec{\tau_{w}}\right|\;dt,$$
where T is the cardiac cycle.

The governing equation for vessel wall deformation is based on the linear momentum conservation principle.
(8)$$\rho_{s}\;\left(\frac{\partial V_{s}}{\partial t}+V_{s}.\nabla V_{s}-g\right)-\nabla \sigma_{s}=0,$$
where s refers the structure domain, $\rho_{s}$ is the density of the solid, V is the velocity field represents the solid displacements ($V_{s}=\frac{\partial u_{s}}{\partial t}$), and σ the Cauchy stress tensor.

For the arbitrary Lagrangian–Eulerian (ALE) framework the conservation equation can be written as follows:(9)$$\rho_{f}\;\left(\frac{\partial V_{f}}{\partial t}+\left[V_{f}-V_{c}\right].\nabla V_{f}\right)-\nabla .\tau_{f}=0,$$
where $\rho_{f}$ is the density of the fluid, $V_{f}$ is the fluid velocity, $V_{c}$ refers the mesh velocity, and $\tau_{f}$ is the fluid shear stress.

Boundary and initial conditions:

To numerically simulate the governing equations, boundary and initial conditions must be known:

#### 2.2.2. Boundary and Initial Conditions

At inlet, the aortic pulsatile pressure was considered at the inlet as shown in Figure 2. This pressure profile was implemented into the numerical simulation using the transient table [40].

At outlet, the mean pressure is considered at the outlet for all numerical simulations. The CA outlet mean pressure is assumed to be 75 mmHg, and the outlet mean pressure for RCA is considered 60 mmHg [46].

For the solid domain, the velocity of the fluid domain is coupled to the elastic structure along with the solid wall boundaries where $V_{f}=V_{c}$.

The pressure and all velocity components are set initially to be zero.

### 2.3. Numerical Simulation

The governing equations are discretized using the finite volume method. Then, an entirely implicit scheme with second-order spatial differences is used to solve the discretized equations. For the coupling of pressure and velocities, the SIMPLE algorithm is employed. A Dell Precision T7500 workstation with an Intel Xeon^®^ processor of 3.75 GHz, 48 cores, and 64 GB installed memory is used to implement parallel computing of the discretized equations.

#### 2.3.1. Grid and Time Step Independent Tests

To select an optimum mesh and time step for the FSI simulation, a grid independence test is conducted for the fluid domain to investigate the influence of grid refinement on the solution. Five different 3D model meshes with cell sizes 0.25, 0.3, 0.4, 0.5, and 0.6 mm correspond to 2,317,041, 1,343,685, 568,121, 292,527, and 170,171 cells, respectively. In these meshes, tetrahedral and prismatic types of elements were used. A cell size of 0.3 mm is selected for the numerical simulation, as the time-averaged wall shear stress (TAWSS) value did not change with respect to 0.25 and 0.3 mm cell size.

#### 2.3.2. Fluid–Structure Interaction (FSI)

FSI methodology is used to capture the deformation and hemodynamics of the vascular vessel. The two-way FSI procedure is coupled by the fluid and structure domains using ANSYS Fluent and Transient structural modules, respectively, and the data is transferred iteratively between them by the implicit scheme. In the two-way FSI coupling procedure, the fluid and structural domains are solved in parallel, which converge together. Accordingly, each fluid and structural domains need to converge before moving to the next time step. Furthermore, the fluid and structure domains are coupled through the arbitrary Lagrangian–Eulerian (ALE) scheme. It captures the fluid and the structure behavior where the structural mesh deforms to adhere to the fluid boundaries, and this movement is transferred to the fluid by adding body forces to the motion equation.

#### 2.3.3. Model Validation

In this study, the developed model was validated using the available experimental data by Shimizu et al. [47]. The measurements of the deformation of a Poly (vinyl alcohol) Hydrogel (PVA-H) stenotic model were used to validate the numerically calculated deformation. Figure 3 compares the current numerically predicted results of the deformation at the narrowest segment of the cross-section and those measured by Shimizu et al. [47]. The PVA-H model was subjected to different pressures from 0 to 100 mmHg. Accordingly, the deformation values were evaluated and compared for all pressures. Based on Figure 3, the current predicted deformation values agree with those of Shimizu et al. [47]. However, the maximum deviation between the current predicted results and the measured experimental values was 15.4%. This variation is mostly related to the data extraction from the study, device accuracy, and experimental measurements and limitations. In addition, it was found that at 20 mmHg, the deformation change at the narrowest segment was approximately 0.106 mm and slightly overpredicted. In the same context, at 50 mmHg, the deformation change at the narrowest segment was 0.165 mm compared to 0.148 mm from the measured data.

## 3. Results and discussion

### 3.1. Wall Shear Stress (WSS)

The WSS induced due to the arterial blood flow system influences the deposition of cholesterol beneath the internal layer of the arteries. Accordingly, the WSS is considered to be an indication of atherosclerotic plaque formation in the vascular system [42]. Furthermore, high shear stresses near the stenotic throat can activate platelets and thus cause thrombosis and complete blockage of the blood flow to the heart or the brain [42]. An exciting piece of evidence from different studies suggests that wall shear stress is associated with the plaque’s development characteristics with low defined as less than 1 Pa and high WSS values greater than 2.5 Pa, which are frequently associated with high-risk plaque features [7,8].

The local allocation of the WSS for the stenotic RCA and CA at the peak systole of the cardiac cycle is presented in Figure 4, Figure 5 and Figure 6. For stenotic RCA—model A, it was found that the maximum value of WSS is located at the stenosis throat, where it reached 508.8 Pa in case of including the arterial compliance compared to 456.7 Pa for the rigid wall assumption, as shown in Figure 4. Additionally, the artery bifurcation is considered as the region of interest where the value of WSS reached 64.4 Pa by considering the wall elasticity and 61.9 Pa for the rigid wall assumption. Furthermore, as shown in Figure 4, it was found that the region of WSS contours for the stenotic RCA changed by including the wall compliance. Accordingly, the high WSS region downstream of the stenosis was smaller for the arterial elasticity effect than the rigid wall assumption, which agrees with Failer et al. [19]. Moreover, the side branch of outlet (2) faces a low WSS of less than 1 Pa, as shown in Figure 3; hence a high risk of plaque initiation appeared to locate in this region.

On the other hand, for the stenotic ICA—model B, it was found that the maximum value of WSS located at the stenosis throat, which reached 116.9 Pa in case of considering the wall elasticity compared to 118.8 Pa for the rigid wall assumption, as shown in Figure 5. Additionally, at the artery bifurcation, the value of WSS reached 16.3 Pa by considering the wall elasticity and 16.2 Pa for the rigid wall assumption. It can be noticed that the values of WSS at the stenotic throat and the artery bifurcation are almost the same by including the arterial wall compliance and the rigid wall assumption, which agrees with De Wilde et al. [48]. Moreover, it was found that the region of contours for the stenotic CA did not change by using the wall compliance compared to the rigid wall assumption.

Moreover, for the stenotic CCA—model C, it was found that the maximum value of WSS located at the stenotic region reached 63.01 Pa in case of considering the wall elasticity compared to 61.52 Pa for the rigid wall assumption, as shown in Figure 6. Additionally, at the artery bifurcation, the value of WSS reached 27.3 Pa by considering the wall elasticity and 25.9 Pa  for the rigid wall assumption. It can be noticed that the values of WSS at the stenotic throat and the artery bifurcation are almost the same by including the arterial wall compliance and the rigid wall assumption which agrees with De Wilde et al. [48]. Moreover, it was found that the region of contours for the stenotic CCA did not change by using the wall compliance compared to the rigid wall assumption.

Furthermore, Table 2 compares the WSS values at the peak systole of the cardiac cycle between the stenotic RCA and CA arteries with FSI and those obtained under rigid wall assumption. Therefore, it can be concluded that considering the wall elasticity is not significant in simulating large arteries such as the carotid artery, as the WSS values were found to be close by including the arterial wall compliance and the rigid wall. However, for the stenotic RCA, it was found that the rigid wall assumption underestimates the WSS values at the stenosis throat by 10.24% compared to including the arterial elasticity. Additionally, it was found that the rigid wall assumption underestimates the WSS values at the RCA bifurcation by 3.8% compared to considering the wall elasticity.

### 3.2. Time-Averaged Wall Shear Stress (TAWSS)

The TAWSS is defined as the mean value of WSS as presented in Equation (7). For the stenotic RCA—model A, the TAWSS values at the stenotic throat were found to be 295.1 Pa in the case of considering the wall elasticity compared to 267.4 Pa for the rigid wall assumption. Additionally, at the artery bifurcation, the TAWSS values were 30.3 Pa in case of including the arterial wall elasticity compared to 28.9 Pa for the rigid wall assumption.

On the other hand, it was found that for the stenotic ICA—model B, the values of TAWSS at the stenosis throat were 56.8 Pa in the case of considering the arterial wall elasticity and 57.3 Pa for the rigid wall. Furthermore, the values of TAWSS at the CA bifurcation were found to be 9 Pa in the case of including the arterial wall docility and 8.5 Pa by using the rigid wall.

Additionally, it was found that for the stenotic CCA—model C, the values of TAWSS at the stenosis throat were 37.9 Pa in the case of considering the arterial wall compliance and 34.65 Pa for the rigid wall. Furthermore, the values of TAWSS at the CA bifurcation were found to be 15.5 Pa in the case of including the arterial wall docility and 14.2 Pa by using the rigid wall.

Regarding the WSS, the TAWSS is known to be an indicator of the formation of plaques in the cardiovascular system. Accordingly, the depositions of plaques beneath the internal layer of the artery influence the artery that appeared to be at risk due to high and low values of TAWSS, which exceed 2.5 Pa and less than 1 Pa.

Furthermore, Table 3 compares the TAWSS values between the stenotic RCA and CA arteries with FSI and rigid wall assumption. From the values of TAWSS, it can be concluded that the vessel segments appear to be at high risk for thrombogenicity and damage to endothelial cells. Moreover, including the arterial wall elasticity is not significant in simulating large arteries such as the carotid artery, as the TAWSS values were close by including the arterial wall elasticity and the wall rigidity assumption. However, for the stenotic RCA, it was found that the rigid wall assumption reduces the TAWSS values at the stenosis throat by 9.39% compared to those obtained by the FSI approach. Additionally, it was found that the wall rigidity assumption underestimates the TAWSS values at the RCA bifurcation of 4.64% compared to wall compliance.

### 3.3. Velocity Field and Streamlines

The streamlines demonstrate the flow velocity direction inside the artery, the stagnation points, and the eddies’ locations.

Figure 7 shows the flow field streamlines for the stenotic RCA—model A. It was found that the peak value of the velocity at the stenotic throat reached 4.77 m/s by including the wall elasticity compared to 4.34 m/s for the rigid wall assumption. This difference is due to increasing the inlet flow rate in the case of the elastic wall. Considering the wall elasticity enlarges the flow area and decreases the flow resistance. Moreover, due to the arterial wall deformation by using the FSI procedure, it was found that in the case of including the wall compliance, all the flows are directed towards the outlet (1). However, for the rigid wall, the flow was distributed between the two outlets. Furthermore, after the stenotic throat, the blood behaves as a jet which affects velocity streamline distribution and the eddies’ existence. Additionally, the WSS is a frictional force, and it is a function of the velocity gradient exerted parallel to the arterial vessel wall that leads to alteration of the endothelial cell leading to plaque development. Accordingly, the internal wall of the vessel at the artery bifurcation faces a jet of flow that increases the WSS, which causes erosion for the arterial wall, especially for stenoses near it, and causes plaques to rupture [49]. Moreover, eddies that exist after the throat area increase the chances of plaque formation and development. Accordingly, these regions with eddies and low velocities affect the stenosis lesion progression, where the deposition of cholesterol and other lipids are beneath the internal layer of the artery.

On the other hand, Figure 8 shows the flow–field streamlines for the stenotic ICA—model B. It was found that the peak value of the velocity at the stenotic throat which reached 2.24 m/s by considering the arterial wall elasticity compared to 2.25 m/s for the rigid wall assumption. It is notable that the maximum velocity in the ICA branch did not change for both the elastic and the rigid wall due to low arterial wall deformation, which will be discussed later. Additionally, it is noticeable that the recirculation zone after the stenotic is larger in case of considering the arterial wall compliance than the rigid wall assumption, which decreases the WSS values, hence increasing the probability of plaque initiation.

Furthermore, after the throat area, these regions with eddies and low velocities affect the stenosis lesion progression, where the deposition of cholesterol and other lipids are beneath the internal layer of the artery.

Moreover, due to the curvature in the ECA, it was found that the velocity is greater than 2.5 m/s. The internal wall of the vessel at the artery ECA faces high values of flow velocities that affected the WSS distribution.

Additionally, Figure 9 shows the flow–field streamlines for the stenotic CCA—model C. It was found that the peak value of the velocity at the stenotic throat reached 1.89 m/s by considering the arterial wall elasticity compared to 1.87 m/s for the rigid wall assumption. Notably, the maximum velocity in the CCA did not change for both the elastic and the rigid wall due to low arterial wall deformation which will be discussed later. Additionally, it is noticeable that the recirculation zone after the stenotic region is more significant in the case of considering the arterial wall compliance than the rigid wall assumption, which decreases the WSS values, hence increasing the probability of plaque initiation. In addition, the peak velocity at the stenotic throat is more prominent in the case of stenotic ICA compared to stenotic CCA, which also appeared in WSS values, which were more remarkable in the case of stenotic ICA than stenotic CCA. Furthermore, after the throat area, these regions with eddies and low velocities affect the stenosis lesion progression, where the deposition of cholesterol and other lipids are beneath the internal layer of the artery.

Moreover, due to the curvature in the ECA, it was found that the velocity is greater than 2.5 m/s. The internal wall of the vessel at the artery ECA faces high values of flow velocities that affected the WSS distribution.

### 3.4. Mass Flow Rate Distribution

The stenotic throats in the vascular system affect the flow rate in the arteries. Hence, the oxygenated blood flow to the heart or the head will change. Therefore, the patient is exposed to high risk due to the reduction in the oxygenated blood flow. Furthermore, the mass flow rate distribution changed significantly with the simulation procedure considering the arterial wall elasticity or the rigid wall assumption.

Regarding the stenotic RCA—model A, it was found that the mass flow rate distribution varied with considering the wall elasticity, as shown in Table 4. At the late diastole of the cardiac cycle, by using the FSI simulation procedure, the inlet mass flow rate is 2.1 g/s compared to 1.94 g/s for the rigid wall assumption. The outlet mass flow is 2.095 g/s from outlet (1) and 0 g/s from outlet (2). For rigid wall, the outlet mass flow rate is 1.727 g/s from outlet (1) and 0.215 g/s from outlet (2). Additionally, at the peak systole of the cardiac cycle, the inlet mass flow rate is 3.508 g/s compared to 3.168 g/s for the rigid wall assumption. The outlet mass flow is 3.508 g/s from outlet (1) and 0 g/s from outlet (2). For rigid wall, the outlet mass flow rate is 2.446 g/s from outlet (1) and 0.716 g/s from outlet (2).

The deviation between the elastic and rigid wall assumption is due to the deformation of the other side of the artery wall of the stenotic throat relative to the plaque’s formation on the wall side. Additionally, it can be noticed that no flow goes out at outlet (2) of the RCA; the resistance increased, which boosted the inlet mass flow rate and the flow velocity directed to outlet (1).

Regarding the CA’s stenotic ICA—model B, Table 5 presents the mass flow rate distribution using the simulation procedure. At the late diastole of the cardiac cycle, the inlet mass flow rate is 19.75 g/s compared to 19.92 g/s for the rigid wall assumption. The outlet mass flow is 9.67 g/s from ICA and 10.08 g/s from ECA. For the rigid wall assumption, the outlet mass flow rate is 9.77 g/s from ICA and 10.14 g/s from ECA. At the peak systole of the cardiac cycle, the inlet mass flow rate is 43.7 g/s compared to 43.63 g/s for the rigid wall assumption. The outlet mass flow is 21.3 g/s from ICA and 22.4 g/s from ECA. For the rigid wall assumption, the outlet mass flow rate is 21.3 g/s from ICA and 22.35 g/s from ECA. It can be noticed that the mass flow rate distribution in the case of elastic to rigid wall assumption is almost insignificant.

Table 6 presents the mass flow rate distribution using the simulation procedure for the CA’s stenotic CCA—model C. At the late diastole of the cardiac cycle, the inlet mass flow rate is 18.89 g/s compared to 19.62 g/s for the rigid wall assumption. The outlet mass flow is 10.2 g/s from ICA and 8.7 g/s from ECA. For the rigid wall assumption, the outlet mass flow rate is 10.6 g/s from ICA and 8.9 g/s from ECA. At the peak systole of the cardiac cycle, the inlet mass flow rate is 40.73 g/s compared to 40.44 g/s for the rigid wall assumption. The outlet mass flow is 21.8 g/s from ICA and 18.9 g/s from ECA. For the rigid wall assumption, the outlet mass flow rate is 21.65 g/s from ICA and 18.8 g/s from ECA. It can be noticed that the mass flow rate distribution in the case of elastic to rigid wall assumption is almost insignificant.

It can be concluded that the numerical simulation considering the elastic wall is necessary for the relatively small diameter vessels, such as the right coronary artery, and can be neglected for the relatively large arteries, such as the carotid artery.

### 3.5. Arterial Wall Deformation

The displacement of the vessel walls significantly affects the flow pattern due to the forces acting on its wall. Radial strain represents the cross-sectional area deformation in terms of the relative radial deformation of the vessel wall to a reference diameter [50]. Fluid flow and solid structure domains are coupled, and their interaction is obtained through the arterial wall strains.

Figure 10 presents the RCA wall radial deformation contours for different times along the cardiac cycle representing the early diastole, peak systole, and late diastole. It was found that the maximum value for the arterial wall radial deformation at the early diastole (0.015 s) reached 0.0717 mm and decreased to 0.00702 mm at 0.322 s, as shown in Figure 10a,b. At the peak systole of the cardiac cycle at 0.38 s, the arterial wall radial deformation’s maximum value was 0.0095 mm at nearly the mid-distance between the artery inlet and the stenotic throat, as shown in Figure 10c. Afterward, Figure 10d,e shows the arterial wall radial displacement at the late diastole of the cardiac cycle which the maximum value decreased from 0.00399 mm at a time 0.5 s of the cardiac cycle to 0.00298 mm at a time 0.74 s of the cardiac cycle.

Figure 11 presents the stenotic ICA—model B wall radial deformation contours for different times along the cardiac cycle at the early diastole, peak systole, and late diastole, defined in Figure 2. It was found that the maximum value for the arterial wall radial displacement at the early diastole (0.02 s) reached 0.0345 mm after the stenotic region of the ICA and increased to 0.0359 mm at a time 0.16 s of the cardiac cycle but located before the artery turn of the ECA branch, as shown in Figure 11a,b. At the peak systole of the cardiac cycle (0.385 s), the maximum value of the arterial wall radial deformation was 0.0219 mm, located after the ICA’s stenotic throat, as shown in Figure 11c. Afterward, Figure 11d shows the arterial wall displacement at the late diastole of the cardiac cycle, in which the maximum value was 0.0108 mm at a time 0.74 s of the cardiac cycle.

Moreover, the stenotic CCA—model C wall radial deformation contours are presented for different times along the cardiac cycle at the early diastole, peak systole, and late diastole, defined in Figure 2. It was found that the maximum value for the arterial wall radial displacement at the early diastole (0.02 s) reached 0.029 mm after the stenotic region of the CCA at the ECA and increased to 0.084 mm at a time 0.16 s of the cardiac cycle but located before the artery turn of the ECA branch as shown in Figure 12a,b. At the peak systole of the cardiac cycle (0.385 s), the maximum value of the arterial wall radial deformation was 0.0427 mm, located after the CCA’s stenotic throat at the ICA, as shown in Figure 12c. Afterward, Figure 12d shows the arterial wall displacement at the late diastole of the cardiac cycle, for which the maximum value was 0.0187 mm at a time 0.74 s of the cardiac cycle.

It can be concluded that the deformation output from the numerical results can be validated with the theoretical wall expansion. The theoretical wall expansion assumption follows Barlow’s formula.
(10)$$H=\frac{P\times D}{2t}$$
where *H* is the hoop stress, *P* is the pressure, *D* is the diameter, and *t* is the wall thickness.

Additionally, the strain can be calculated using the following equation
(11)ϵ=H/E 
where ϵ is the strain and *E* is the Young modulus of elasticity.

For stenotic RCA—model A at the stenotic throat, with pressure = 5406 Pa, diameter = 3.77 mm, thickness = 0.55 mm, and the Young modulus of elasticity = 1.08 MPa, hence the strain = 0.0171 and in diameter = 0.064 mm which agrees with Figure 10. Additionally, for stenotic ICA—model B at the stenotic throat, with pressure = 450 Pa, diameter = 7.82 mm, thickness = 0.66 mm, and the Young modulus of elasticity = 1.08 MPa, hence the strain = 0.00247 and in diameter = 0.0193 mm which agrees with Figure 11. In the same context, for stenotic CCA—model C at the stenotic throat, with pressure = 5531 Pa, diameter = 7.82 mm, thickness = 0.66 mm, and the Young modulus of elasticity = 1.08 MPa, hence the strain = 0.0303 and in diameter = 0.237 mm which agrees with Figure 12. Moreover, it is well known that the deformation occurred due to the pressure forces near the arterial wall. However, Figure 7, Figure 8 and Figure 9 show the velocity streamlines inside the RCA and CA geometries by using the wall compliance and rigidity assumption to compare the effect of using the rigid wall assumption on the hemodynamics. Furthermore, Figure 10, Figure 11 and Figure 12 show the deformation that occurred in the elastic wall geometries due to the presence of pressure forces on the arterial walls. Additionally, Figure 13 shows the pressure contours along the studied geometries at the peak systole of the cardiac cycle.

## 4. Conclusions

The occurrence of stenosis significantly affects the blood flow dynamic factors, such as WSS, velocity distribution, and arterial wall deformation. Hence, hemodynamics can be used as a threshold to explore the influence of using the rigid wall assumption along the coronary and the carotid arteries. Therefore, comprehensive blood flow models were developed and numerically simulated to predict the blood flow dynamic factors.

It can be concluded that considering the arterial wall elasticity is unnecessary in simulating large arteries such as the carotid artery. It was found that the hemodynamics for the stenotic carotid artery (CA) were almost the same when using the elastic and rigid wall assumptions. The WSS at the stenosis throat reached 116.9 Pa, considering the wall elasticity compared to 118.8 Pa for the rigid wall assumption. Additionally, at the artery bifurcation, the value of WSS reached 16.3 Pa by considering the wall elasticity and 16.2 Pa for the rigid wall assumption. Hence, assuming the elastic wall assumption for relatively large arteries such as the CA can be neglected to save the computational cost.

However, for the stenotic right coronary artery (RCA), it was found that adopting the rigid wall assumption underestimates the WSS values at the stenosis throat by 10.24% compared to simulating the blood flow considering wall elasticity. Additionally, it was found that rigid wall assumption underestimates the WSS values at the RCA bifurcation by 3.8% compared to considering wall compliance.

Accordingly, the blood flow dynamic factors were significantly influenced by using the elastic wall assumption for relatively small diameters of the blood vessels, such as RCA.Additionally, the rigid wall assumption is plausible in flow modeling for relatively large diameters such as the carotid artery.

## Figures and Tables

**Figure 1 bioengineering-09-00708-f001:**
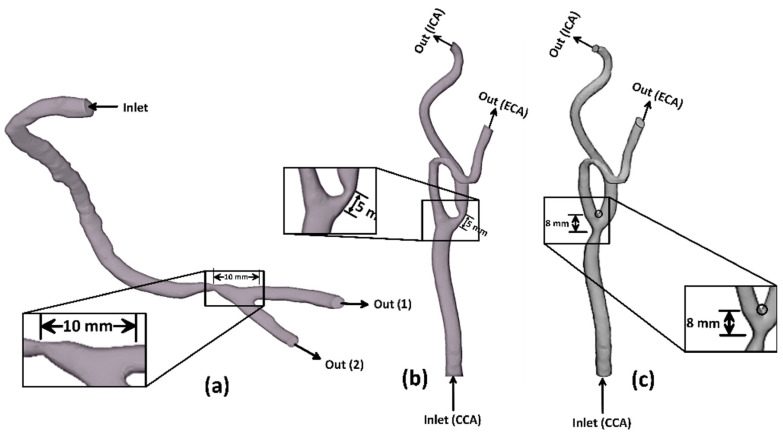
Physical model schematic diagram: (**a**) stenotic RCA—model A, (**b**) stenotic ICA—model B, and (**c**) stenotic CCA—model C.

**Figure 2 bioengineering-09-00708-f002:**
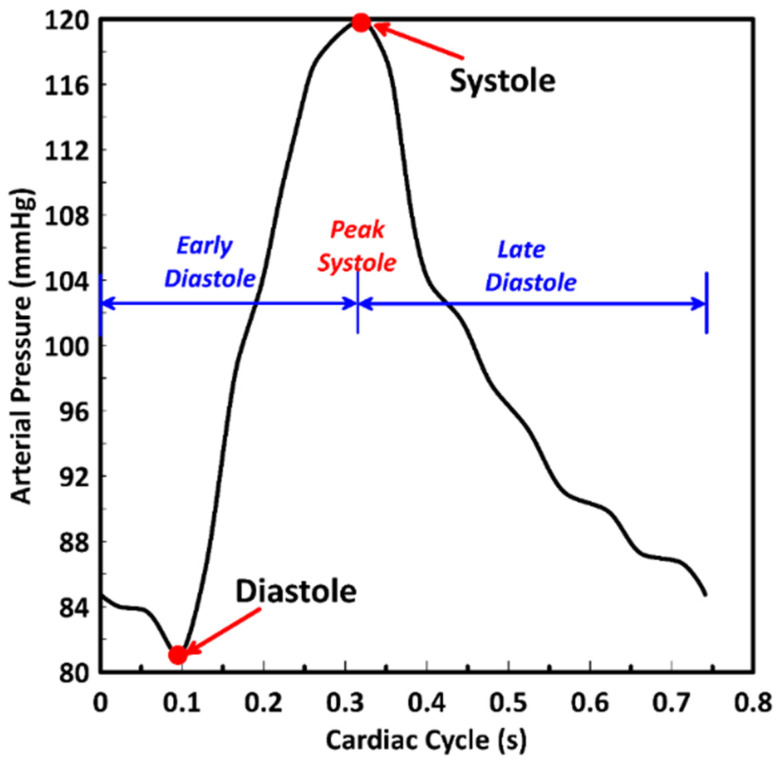
Inlet pressure distribution along the cardiac cycle represents the early diastole, peak systole, and late diastole periods.

**Figure 3 bioengineering-09-00708-f003:**
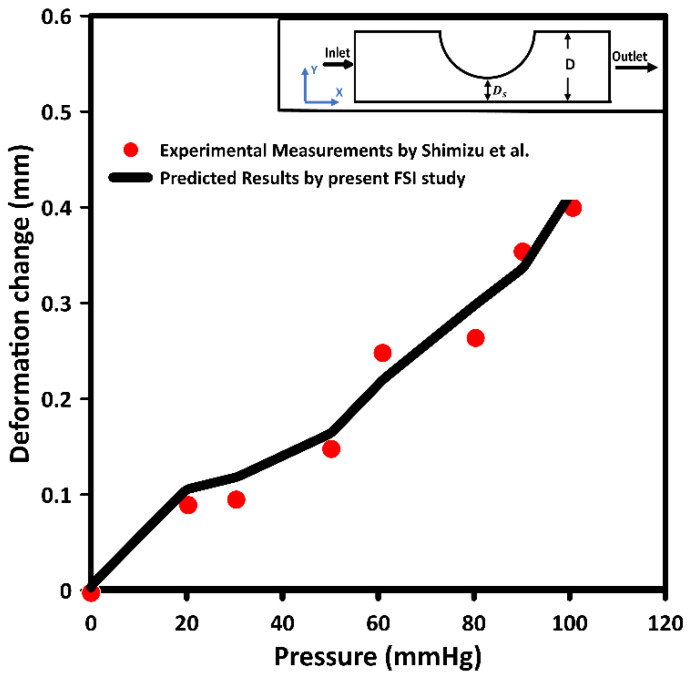
Validation of current calculated deformation changes with those of Shimizu et al. [47] at the narrowest segment of a stenotic model.

**Figure 4 bioengineering-09-00708-f004:**
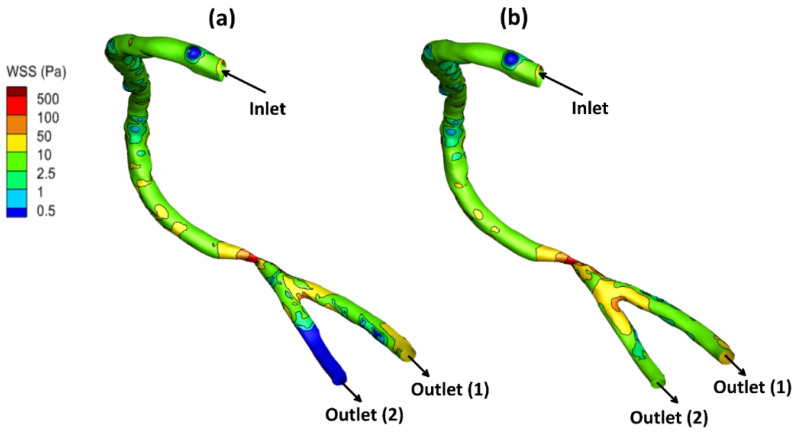
WSS contours for stenotic RCA (model A) at the peak systole of the cardiac cycle: (**a**) elastic wall, (**b**) rigid wall.

**Figure 5 bioengineering-09-00708-f005:**
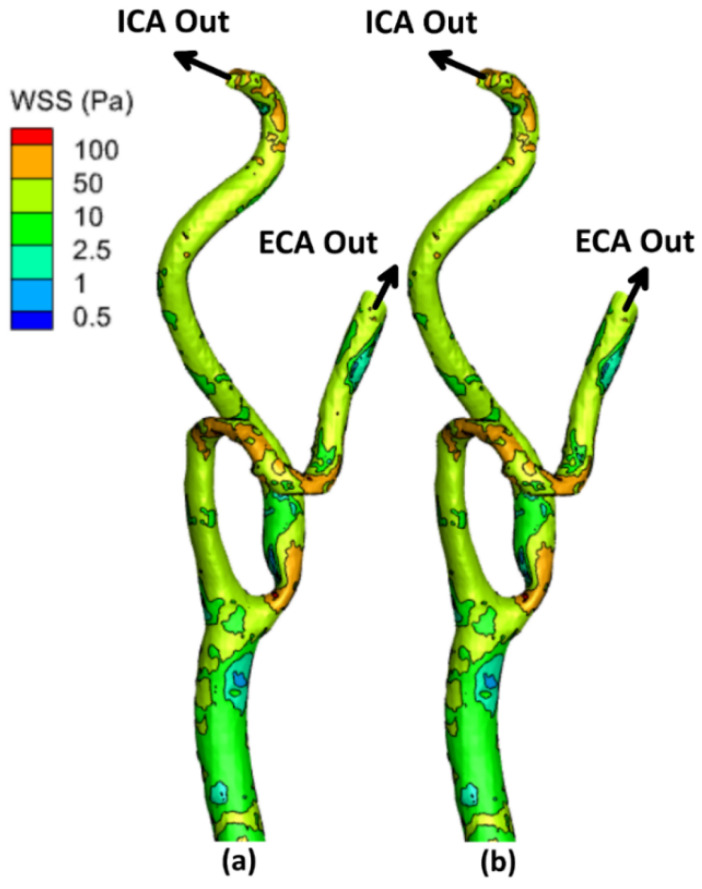
WSS contours for stenotic ICA (model B) of the CA at the peak systole of the cardiac cycle: (**a**) elastic wall, (**b**) rigid wall.

**Figure 6 bioengineering-09-00708-f006:**
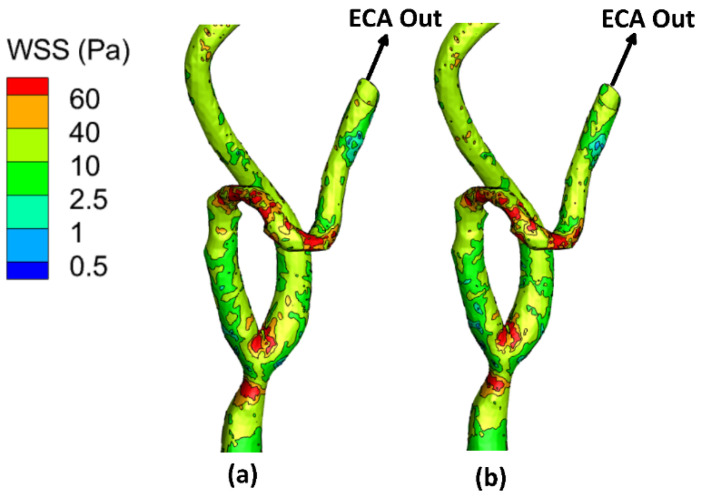
WSS contours for stenotic CCA (model C) of the CA at the peak systole of the cardiac cycle: (**a)** elastic wall, (**b**) rigid wall.

**Figure 7 bioengineering-09-00708-f007:**
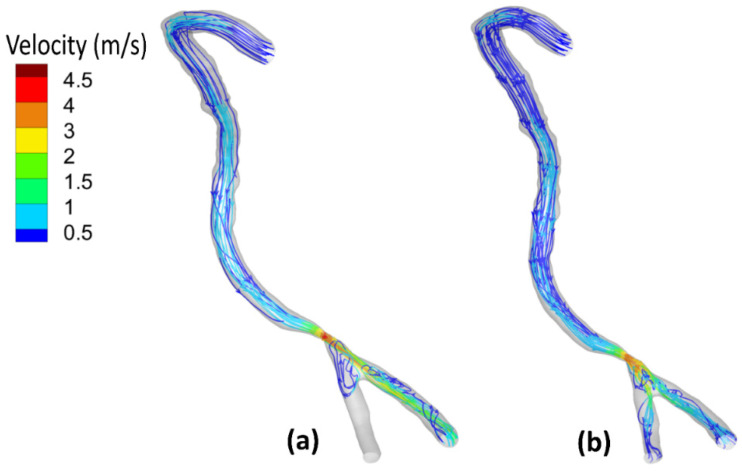
Streamlines at the peak systole for stenotic RCA (model A): (**a**) elastic wall, (**b**) rigid wall.

**Figure 8 bioengineering-09-00708-f008:**
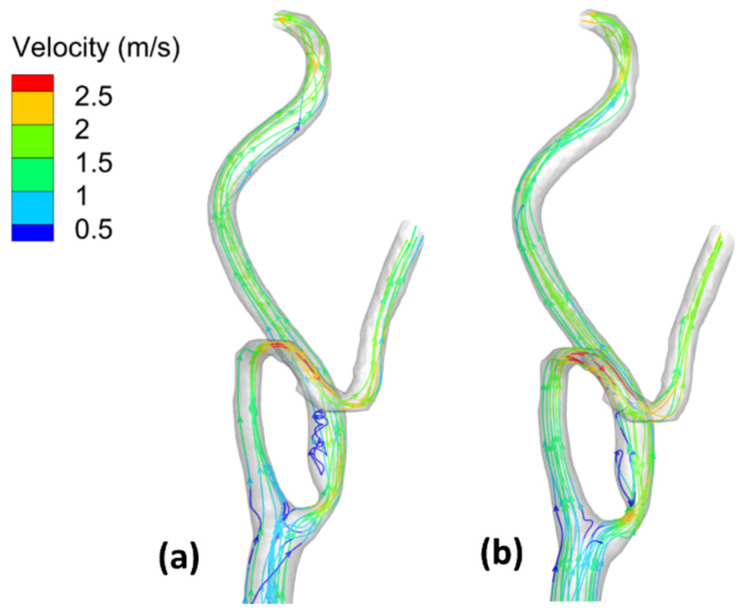
Streamlines at the peak systole for stenotic ICA (model B) of the CA: (**a**) elastic wall, (**b**) rigid wall.

**Figure 9 bioengineering-09-00708-f009:**
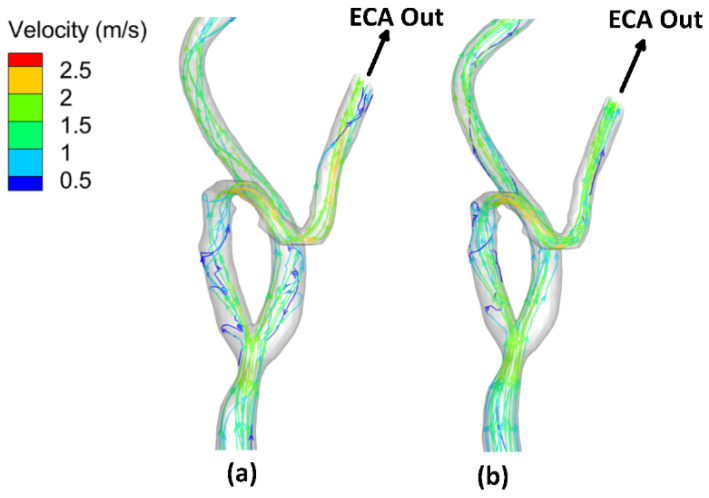
Streamlines at the peak systole for stenotic CCA (model C) of the CA: (**a**) elastic wall, (**b**) rigid wall.

**Figure 10 bioengineering-09-00708-f010:**
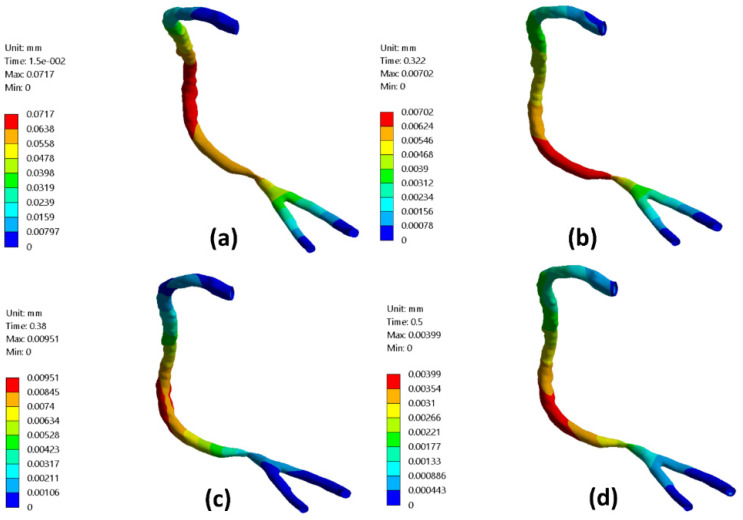
RCA—model A deformation contours for different times of the cardiac cycle: (**a**) t = 0.015 s, (**b**) t = 0.32 s, (**c**) t = 0.38 s, (**d**) t = 0.5 s, and (**e**) t = 0.74 s.

**Figure 11 bioengineering-09-00708-f011:**
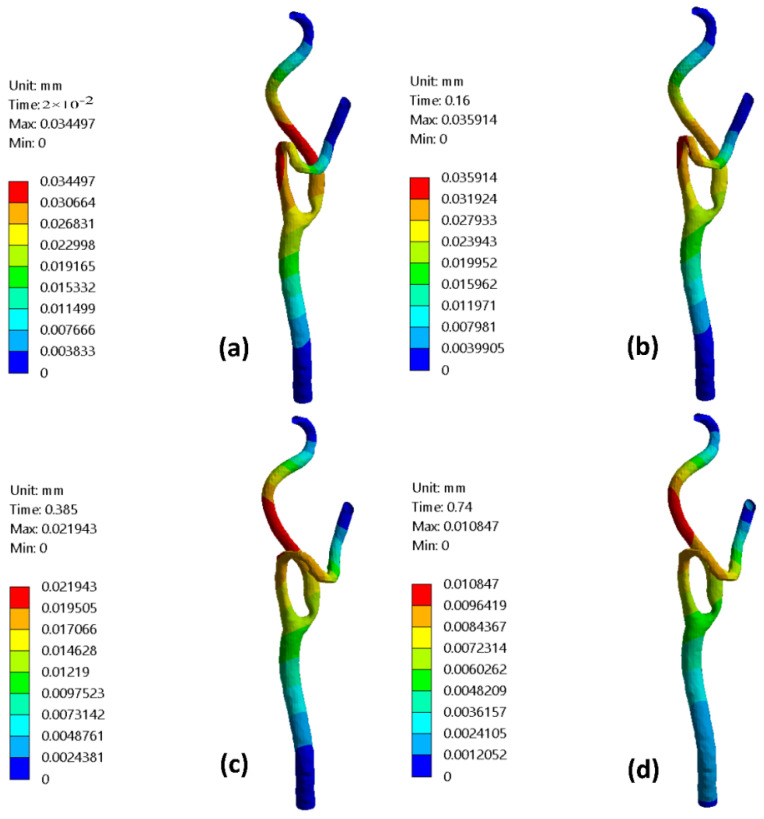
CA—model B deformation contours for different times of the cardiac cycle: (**a**) t = 0.02 s, (**b**) t = 0.16 s, (**c**) t = 0.385 s, and (**d**) t = 0.74 s.

**Figure 12 bioengineering-09-00708-f012:**
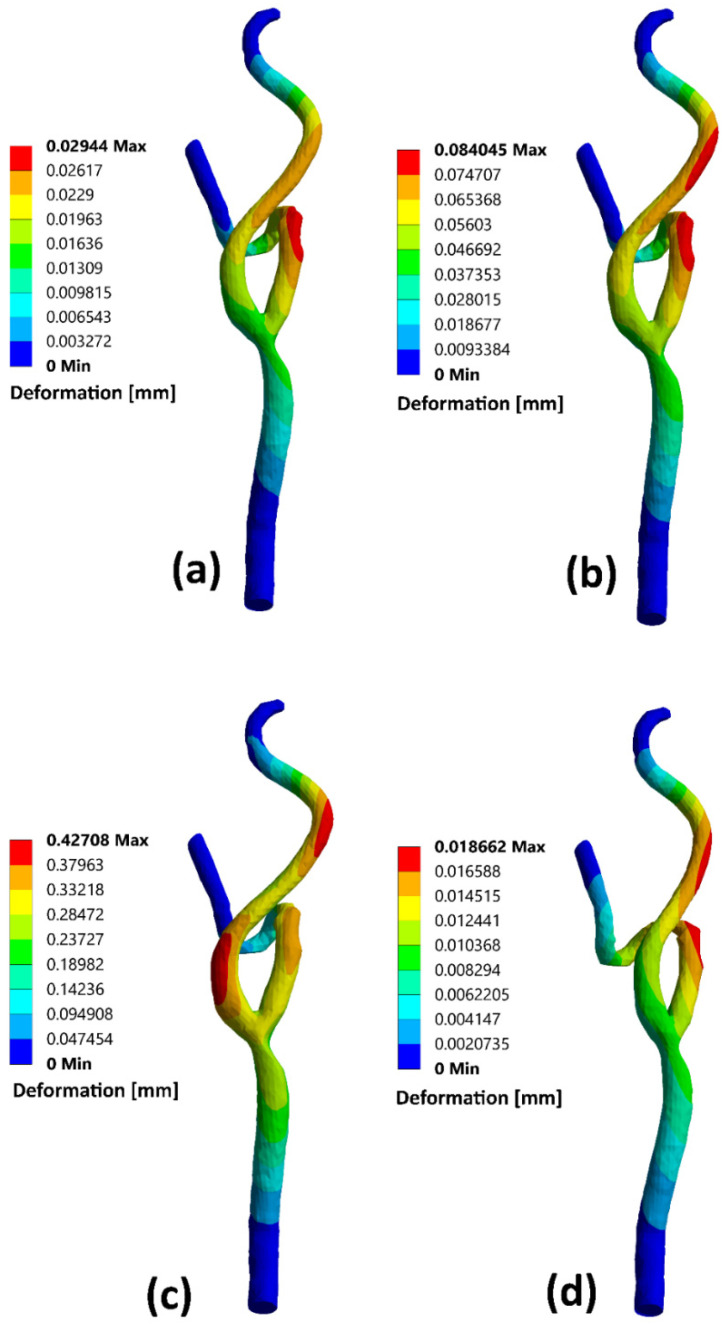
CA—model C deformation contours for different times of the cardiac cycle: (**a**) t = 0.02 s, (**b**) t = 0.16 s, (**c**) t = 0.385 s, and (**d**) t = 0.74 s.

**Figure 13 bioengineering-09-00708-f013:**
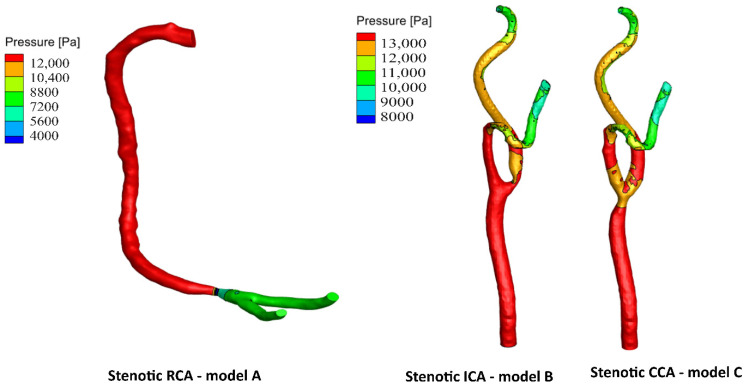
Pressure distribution for the wall compliance models at the peak systole of the cardiac cycle.

**Table 1 bioengineering-09-00708-t001:** Physical model dimensions.

	Nominal Hydraulic Diameter			Throat Diameter	Wall Thickness
	Main Branch	Outlet (1)	Outlet (2)		
Stenotic RCA—model A	3.77 mm	2.4 mm	1.9 mm	1.51 mm	0.55 mm
	**CCA**	**ECA**	**ICA**		
Stenotic ICA—model B	7.82 mm	4.52 mm	6.41 mm	3.13 mm	0.66 mm
Stenotic CCA—model C	7.82 mm	4.52 mm	6.41 mm	3.13 mm	0.66 mm

**Table 2 bioengineering-09-00708-t002:** WSS for the stenotic RCA and CA at the peak systole of the cardiac cycle elastic and rigid wall.

WSS (Pa)				
	Stenosis Throat		Artery Bifurcation	
	Elastic Wall	Rigid Wall	Elastic Wall	Rigid Wall
Stenotic RCA—model A	508.8	456.7	64.4	61.9
Stenotic ICA—model B	116.9	118.8	16.3	16.2
Stenotic CCA—model C	63.01	61.5	27.3	25.9

**Table 3 bioengineering-09-00708-t003:** TAWSS for the stenotic RCA and CA for elastic and rigid wall.

TAWSS (Pa)				
	Stenosis Throat		Artery Bifurcation	
	Elastic Wall	Rigid Wall	Elastic Wall	Rigid Wall
Stenotic RCA—model A	295.1	267.4	30.3	28.9
Stenotic ICA—model B	56.8	57.3	9	8.5
Stenotic CCA—model C	37.9	34.65	15.5	14.2

**Table 4 bioengineering-09-00708-t004:** RCA (model A) mass flow rate distribution at the late diastole and peak systole of the cardiac cycle for elastic and rigid wall.

Flow Rate at Late Diastole (g/s)			
	Inlet	Outlet (1)	Outlet (2)
Elastic wall (FSI)	2.095	2.095	0
Rigid wall (CFD)	1.94	1.727	0.215
**Flow Rate at Peak Systole (g/s)**			
	Inlet	Outlet (1)	Outlet (2)
Elastic wall (FSI)	3.508	3.508	0
Rigid wall (CFD)	3.163	2.446	0.716

**Table 5 bioengineering-09-00708-t005:** Stenotic ICA (model B) mass flow rate distribution at the late diastole and peak systole of the cardiac cycle for elastic and rigid wall.

Flow Rate at Late Diastole (g/s)			
	Inlet	ICA	ECA
Elastic wall (FSI)	19.75	9.671	10.08
Rigid wall (CFD)	19.92	9.77	10.14
**Flow Rate at Peak Systole (g/s)**			
	Inlet	ICA	ECA
Elastic wall (FSI)	43.7	21.304	22.4
Rigid wall (CFD)	43.63	21.3	22.35

**Table 6 bioengineering-09-00708-t006:** Stenotic CCA (model C) mass flow rate distribution at the late diastole and peak systole of the cardiac cycle for elastic and rigid wall.

Flow Rate at Late Diastole (g/s)			
	Inlet	ICA	ECA
Elastic wall (FSI)	18.89	10.2	8.7
Rigid wall (CFD)	19.62	10.6	8.9
**Flow Rate at Peak Systole (g/s)**			
	Inlet	ICA	ECA
Elastic wall (FSI)	40.73	21.8	18.9
Rigid wall (CFD)	40.44	21.65	18.8

## Data Availability

The data supporting the findings of this study are available from the corresponding author upon reasonable request.

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
