# Peer review of "Influence of Rigid–Elastic Artery Wall of Carotid and Coronary Stenosis on Hemodynamics"

_bioengineering, 2022, doi:10.3390/bioengineering9110708_

Round 1

Reviewer 1 Report

Dear Authors 

I have some minor revisions:

In the abstract you must provide spelling of acronyms first time you report it

Methods: Specific how many patients have you detected.

Do you consider if a very calcified plaque could influence your analysis at the level of bifurcation? In these cases somewhere it is complicate to seare arterial wall from the plaque. Please comment

Author Response

The authors greatly appreciate the editor and reviewers’ valuable comments. All comments were carefully considered in the revised version of the manuscript.

Response to reviewer # 1

  • In the abstract you must provide spelling of acronyms first time you report it
    Response:

The authors apologize for the inconvenience, in the revised version of the manuscript, the abstract is modified to the reviewer comment.

  • Methods: Specific how many patients have you detected.

Response:

This sentence is added to the manuscript at section 2.1 physical model

The anatomic data of two healthy patient-specific geometries were detected with the carotid artery (CA) and right coronary artery (RCA), which were obtained from a clinically indicated Computed Tomography (CT) angiogram.

  • Do you consider if a very calcified plaque could influence your analysis at the level of bifurcation? In these cases somewhere it is complicate to severe arterial wall from the plaque. Please comment

Response:

Further studies will be conducted with more stenoses’ severities, locations, and shapes in order to investigate the assessment of the rigid wall assumption on the hemodynamics.

Reviewer 2 Report

Review comments for Bioengineering-1982307

Influence of rigid-elastic artery wall of carotid and coronary stenosis on hemodynamics

M. Albadawi, Y. Abuouf, S. Elsagheer, H. Sekiguchi, S. Ookawara, M. Ahmed

An interesting numerical study of the influence of elastic walls on the hemodynamics of carotid and coronary stenosed arteries is presented. The authors should correct language errors, improve the narrative and clarity of the text and address a few technical deficiencies. Improving suggestions are listed below:

01. Lines 2-3: A more representative title would be “Numerical study of the influence of elastic walls on the hemodynamics of carotid and coronary stenosed arteries”.

02. L24: Avoid using numerals in the Abstract. Better to write “the maximum value of wall shear stress (WSS) for the FSI case is higher than that for the rigid wall”.

03. L25-26: Better to write “On the other hand, for the stenotic carotid artery, it was found that the maximum value of WSS for the FSI case is lower than that for the rigid wall”.

04. L27 & 29: Use percentages instead of absolute values for the wall deformations.

05. L30: Delete the words “that a”.

06. L36-118: Correct language errors and improve the narrative and clarity of the Introduction.

07. L46: Better to write “It is” instead of “It’s”.

08. L48-49: Better to write “accumulation” instead of “accumulate”.

09. L55: Write “an increase of WSS value is observed” instead of “it is observed an increase of WSS value”.

10.L73-74: Rephrase this sentence to improve its narrative and clarity.

11. L78: Write “High-complexity” instead of “high- complexity”.

12.L83-84: Rephrase this sentence to improve its narrative and clarity.

13. L88: Better to write “vessel mechanical properties” instead of “vessel's mechanical properties”.

14. L98: Delete the word “morphologically” as redundant.

15. L119: Use as heading “2. Problem definition and modelling”.

16. L120-148: Correct language errors and improve the narrative and clarity of section 2.1.

17. L153-177: Correct language errors and improve the narrative and clarity of section 2.2.

18. L179-213: Correct language errors and improve the narrative and clarity of section 2.2.1.

19. L214: Use as section heading “2.2.2. Boundary and initial conditions” and delete the heading “2.2.3 Initial conditions” in line 226 as unnecessary.

20. L229-261: Correct language errors and improve the narrative and clarity of section 2.3.

21. L265: Better to write “3. Results and discussion”.

22. L266-275: This text could be omitted as unnecessary.

23. L277-335: Correct language errors and improve the narrative and clarity of section 3.1.

24. L340-370: Correct language errors and improve the narrative and clarity of section 3.2.

25. L374-430: Correct language errors and improve the narrative and clarity of section 3.3.

26. L435-486: Correct language errors and improve the narrative and clarity of section 3.4.

27. L491-573: Correct language errors and improve the narrative and clarity of section 3.5.

28. L540-565: Correct language errors and improve the narrative and clarity of the Conclusions. It would also be good to avoid using absolute values in the Conclusions by using percentages wherever possible.

29. Make sure that references are cited and listed complete, consistently, correctly and according to the standards of the Journal.

Reviewer 3 Report

Thank you for submitting such an interesting paper. My comments follow

Please define the reduction of hydraulic diameter specifying nominal and reduced diameter in line 140. For example reducing from 3 mm to 3*(1-0.6)=1.2 mm reduces the cross section from 7.07 mm^2 to 1.13 mm^2 ??? Increasing velocity by 6.25 and shear stress by 6.25/(1-0.6)=15.625??? How is affected Reynolds in these regions?

I suggest you provide values of diameters and thickness from line 142 in a table with reference to positions inside figure 1.

Please estimate theoretical wall expansion using this assumption. Hoopstress is pressure * diameter /(thickness * 2), strain is hoop stress / E and this is related to increment of diameter. Example for increase of 44 mmHg=5333 Pa=0.005333 MPa, diameter 3mm and thickness 0.66mm and E=1.08MPa we obtain 0.005333 MPa*3mm/(2*0.66 mm * 1.08 MPa)=0.011 = 1.12% or in diameter an increase of 3*0.011=0.033 mm in agreement with figure 10.

Please change figure 3 legend to clarify if measurements are experimental results from Shimizu and predicted results are FSI results from current research.

Please provide a theoretical estimation of wall shear stress and pressure drop due to shear stress*length/diameter. How is this value compare to having the average pressure at outlet in simulation?

Please provide a section cut with velocity profile to check velocity profile shape for non-newtonian fluid.

Please make a comment on mesh size and refinement near artery walls.

What are the physics behind the effect of increasing diameter, decreasing velocity and decreasing shear stress compared to contradictory results shown in table 1? Why is increasing flow rate in elastic artery?

Please relate velocities, pressure and deformation figure 7 with 10 and figure 8 with 11... 

Author Response

The authors greatly appreciate the reviewer’s valuable comments. All comments were carefully considered in the revised version of the manuscript.

Reviewer 4 Report

The article investigates blood flow coupled with arterial wall response using  the Fluid Structure Interaction (FSI) procedure as comprehensive realistic approach. Article is well-written and has a potential and my comments are as follows: 

First of all, please the high similarity existing with the other article in both the introduction and discussion.

 Please add more details about the FSI for the common readers to follow in the introduction.

What do you mean by “patient-specific arteries” in the last paragraph in the introduction? That is, why not add the specific arteries’ names indented in the article?

Try to enhance the resolution of Figure 1.

 Punctuation is missing after equations and should be taken care of in the revised version.

What are the special cases to the proposed problem?

 Add the core findings at the end of the abstract.

Only preferably, add a Conclusions section to be written in a brief bullet points for better understanding to the readers to the core findings of the investigation.

For enhancing the introduction section with the new publications, old references may be replaced with new ones such as:

Biomedical simulations of nanoparticles drug delivery to blood hemodynamics in diseased organs: Synovitis problem

Author Response

The authors greatly appreciate reviewer’s valuable comments. All comments were carefully considered in the revised version of the manuscript.

  • First of all, please the high similarity existing with the other article in both the introduction and discussion.

Response:

The authors apologize for the inconvenience, in the revised version of the manuscript, the authors did their best to decrease the plagiarism percentage for this paper to 6% with yellow highlighting, although this published article is also the authors’ manuscript.

  • Please add more details about the FSI for the common readers to follow in the introduction.

Response:

The authors greatly appreciate the reviewer’s valuable comment. This Paragraph is added to the manuscript in the introduction section (lines 90-92) and with green highlighting

FSI is a Multiphysics coupling of fluid dynamics and structural mechanics regulations. This phenomenon, which can be steady or oscillatory, is characterized by interactions between a deformable or moving structure and a surrounding or interior fluid flow.

  • What do you mean by “patient-specific arteries” in the last paragraph in the introduction? That is, why not add the specific arteries’ names indented in the article?

Response:

The authors apologize for the inconvenience, the patient specific arteries are the geometries which investigated in this study, and they are carotid and coronary patient specific geometries. This paragraph is modified in the revised version of the manuscript.

  • Try to enhance the resolution of Figure 1.

Response:

The authors apologize for the inconvenience, the resolution of Figure 1 is modified to enhance the resolution.

  • Punctuation is missing after equations and should be taken care of in the revised version.

Response:

The authors greatly appreciate this valuable comment. The punctuation is added to the revised version.

  • What are the special cases to the proposed problem?

Response:

The anatomic data of two healthy patient-specific geometries were detected with the carotid artery (CA) and right coronary artery (RCA), which were obtained from a clinically indicated Computed Tomography (CT) angiogram. Moreover, eccentric stenosis is imported to the RCA at 10 mm before the bifurcation (model A), as shown in Figure 1-a. Additionally, eccentric stenosis is imported to the healthy geometry of the CA at the internal carotid artery (ICA) 5 mm downstream of the artery bifurcation (model B), as shown in Figure 1-b. Also, eccentric stenosis is imported to the CA at the Common Carotid Artery (CCA) 8 mm before the bifurcation (model C), as shown in Figure 1-c. At each location, a reduction percentage in the hydraulic diameter is 60%. Further studies will be conducted with more stenoses’ severities, locations, and shapes in order to investigate the assessment of the rigid wall assumption on the hemodynamics.

  • Add the core findings at the end of the abstract.

Response:

The authors apologize for the inconvenience, the following sentences are added to the revised version of the manuscript. Findings indicate slight differences in results for large diameter arteries such as the carotid artery. Accordingly, the rigid wall assumption is plausible in flow modeling for relatively large diameters such as the carotid artery. Additionally, The FSI approach is essential in flow modeling in small diameters.

  • Only preferably, add a Conclusions section to be written in a brief bullet point for better understanding to the readers to the core findings of the investigation.

Response:

The authors greatly appreciate the reviewer’s valuable comment. This Paragraph is added to the manuscript in the conclusion section (lines 570-574) and with green highlighting

  • Accordingly, the blood flow dynamic factors were significantly influenced by using the elastic wall assumption for relatively small diameters of the blood vessels, such as RCA.
  • Additionally, the rigid wall assumption is plausible in flow modeling for relatively large diameters such as the carotid artery.

  • For enhancing the introduction section with the new publications, old references may be replaced with new ones such as: Biomedical simulations of nanoparticles drug delivery to blood hemodynamics in diseased organs: Synovitis problem

Response:

The authors greatly appreciate this valuable comment. This article is added to the manuscript in the introduction section (lines 71-73) and with green highlighting.

Mekheimer et al. [14] presented numerically that mixing the blood with the synovial fluid can change the rheological properties of the blood and the mechanical characteristics of the formed stenosis.

Round 2

Reviewer 4 Report

Authors have done the required amendments and article is ready for publication.